# VoiceNoNG: High-Quality Speech Editing Model without Hallucinations

## Abstract

Currently, most advanced speech editing models are based on either neural codec language models (NCLM) (e.g., VoiceCraft) or diffusion models (e.g., Voicebox). Although NCLM can generate higher quality speech compared to diffusion models, it suffers from a higher word error rate (WER) (Peng et al., 2024), calculated by comparing the transcribed text to the input text. We identify that this higher WER is due to attention errors (hallucinations), which make it difficult for NCLM to accurately follow the target transcription. To maintain speech quality and address the hallucination issue, we introduce VoiceNoNG, which combines the strengths of both model frameworks. VoiceNoNG utilizes a latent flow-matching framework to model the pre-quantization features of a neural codec. The vector quantizer in the neural codec implicitly converts the regression problem into a token classification task similar to NCLM. We empirically verified that this transformation is crucial for enhancing the performance and robustness of the speech generative model. This simple modification enables VoiceNoNG to achieve state-of-the-art performance in both objective and subjective evaluations. Lastly, to mitigate the potential risks posed by the speech editing model, we examine the performance of the Deepfake detector in a new and challenging practical scenario. Audio examples can be found on the demo page: `https://anonymous.4open.science/w/NoNG-8004/`

## 1 Introduction

In recent years, speech editing technology has rapidly gained prominence, offering substantial benefits across various domains. This technology enables users to seamlessly modify and enhance audio recordings (Morrison et al., 2024), addressing issues such as slips of the tongue, mispronunciations, or transient background noise. Consequently, YouTubers and filmmakers can edit their speeches without the need for re-recording, significantly boosting productivity and reducing costs. However, like many technological advancements, the rise of speech editing also presents notable risks. The same capabilities that make speech editing valuable for legitimate purposes can be exploited to create sophisticated Deepfake audio (NBC News, 2024), posing serious threats to security, privacy, and public trust. As these tools become more accessible, malicious actors could use them to fabricate convincing audio for disinformation, fraud, or other harmful activities, making it crucial to develop and advance speech Deepfake detection technologies. The research community is increasingly focused on building robust detection systems to identify manipulated speech (Zhang et al., 2021b; 2022; Liu et al., 2024; Zhang et al., 2024; Pan et al., 2024), ensuring that the benefits of speech editing can be enjoyed without compromising safety and security. Therefore, while speech editing represents a significant leap forward in digital communication, it also underscores the need for advanced detection methods to mitigate its potential dangers.

In this study, we first propose a state-of-the-art (SOTA) speech editing model, VoiceNoNG. To avoid malicious applications, we also study the performance of the Deepfake detector in a practical new challenging scenario. An introduction to these two topics is as follows:

### 1.1 Speech editing techniques

In the early stages of speech editing, the process primarily involved cut-and-paste techniques to replace the speech segments that needed to be changed with the target speech segments. However,

this approach often resulted in unnatural intonation and noticeable boundaries. Voco (Jin et al., 2017) addressed this issue with voice conversion to make the voice even closer to the original. Morrison et al. (2021) performed speech editing under a single-speaker scenario to avoid the need for voice conversion, which uses an utterance-level text-to-speech (TTS) system for context-aware synthesis, followed by pitch-shifting, duration-stretching, and finally eliminating speech manipulation artifacts by HiFi-GAN vocoding (Kong et al., 2020). Both of the above methods allow users to manually fine-tune pitch, amplitude, and duration through a user interface to ensure natural transitions at the boundaries.

In recent years, as speech synthesis technology has become more advanced, methods for speech editing have generally been evolved into two main approaches: non-autoregressive speech infilling (inpainting) (Borsos et al., 2022; Bai et al., 2022; Le et al., 2024; Jiang et al., 2023; Alexos & Baldi, 2024; Ruiz et al., 2024), and autoregressive causal generation (Tan et al., 2021; Wang et al., 2024; Peng et al., 2024).

For the training of non-autoregressive model, the models take masked audio along with its transcript as input, and the goal is to recover the masked segments. During speech editing, the segment to be modified is simply masked with a new target transcript, and the corresponding edited speech is generated. Non-autoregressive speech editing models can be further divided based on whether explicit text-audio alignment is required. Models like A3T (Bai et al., 2022), Voicebox (Le et al., 2024), and FluentSpeech (Jiang et al., 2023) require explicit alignment, while models such as SpeechPainter (Borsos et al., 2022), AttentionStitch (Alexos & Baldi, 2024), and Mapache (Ruiz et al., 2024) implicitly learn the alignment through their attention mechanisms.

For autoregressive models, the challenge lies in how to condition the causal generation on both past and future contexts. EditSpeech (Tan et al., 2021) adopts an approach similar to non-autoregressive methods, with the key difference being that masked generation is achieved through two decoders: a forward decoder and a backward decoder. SpeechX (Wang et al., 2024) and VoiceCraft (Peng et al., 2024) fall under the Neural Codec Language Model (NCLM) category. These models first convert audio into discrete tokens by a neural codec, which are then trained as NCLMs through a next-token prediction task. The target text and context information are provided through prompting to help the model learn speech editing. The predicted tokens are then converted to a waveform using the neural codec decoder.

Among these speech editing models, Voicebox (Le et al., 2024) and VoiceCraft (Peng et al., 2024) are the two most famous ones, and our proposed editing method, VoiceNoNG, is based on the combination of these two methods. An overview of the input and output of current speech editing models are shown in Figure 1.

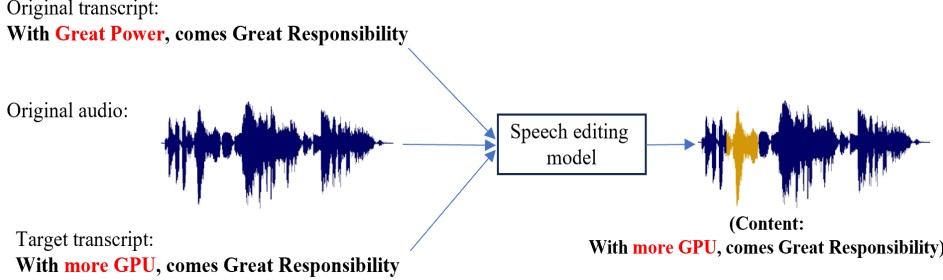

Figure 1: The input and output of speech editing models.

## 1.2 DEEPFAKE SPEECH DETECTION AND LOCALIZATION

To mitigate the potential harm caused by synthetic speech, challenges like ASVspoof (Kinnunen et al., 2017; Todisco et al., 2019; Yamagishi et al., 2021) and ADD (Yi et al., 2023) have been launched to promote defensive research. These primarily address fully spoofed scenarios, where the entire audio is generated through methods like text-to-speech or voice conversion. However,

generating an entire speech may not always be necessary. If attackers have access to real samples with significant overlap with the target transcript, they only need to manipulate short segments using speech editing models. Compared to the conventional fully spoofed scenario, detecting and localizing (Yi et al., 2021; Zhang et al., 2021a; Zhang & Sim, 2022) partial spoof scenarios is more challenging.

Additionally, given the impressive compression capabilities of neural codecs, it is anticipated that they will become a new standard for audio formats (e.g., mp3). For a Deepfake detector, it is insufficient to simply distinguish between real and codec-resynthesized audio (Wu et al., 2024; Xie et al., 2024). A practical new threat to consider in partial spoof scenarios is differentiating between resynthesized and edited speech (Le et al., 2024). In this context, the detector cannot identify edited speech merely by recognizing specific patterns or artifacts generated by the codec. More details will be discussed in Section 3.2.

In summary, this paper makes several significant contributions:

1) We identify the **hallucination-like** problem in the autoregressive-based VoiceCraft model, which makes it difficult to follow the target transcription accurately. This leads to a higher word error rate (WER), calculated by comparing the transcribed text to the target transcription.

2) We propose VoiceNoNG, a speech editing model that combines the strengths of both Voicebox and VoiceCraft, achieving **SOTA** performance in both objective and subjective evaluations.

3) We find that the **vector quantizer** module in the neural codec is crucial for developing a robust speech editing model.

4) We extend the partial spoof detection setting to a more practical scenario. A new speech dataset edited by VoiceNoNG will be released to support defensive research.

## 2 PROPOSED VOICENONG

As mentioned in the previous section, for the edited part to sound natural, the speaker characteristics and background audio (e.g., noise, music, etc.) need to be consistent with the context. In this section, we first discuss the drawbacks of Voicebox and VoiceCraft, and then explain how we address these issues (Voicebox and VoiceCraft can be considered representative speech editing models based on diffusion models and neural codec language models (NCLM), respectively).

Voicebox is a cutting-edge generative speech model based on a non-autoregressive flow-matching approach. Its flow-matching framework allows it to directly learn to transform noise distributions into audio data distributions, leading to more stable training, better generalization, and faster sample generation compared to traditional diffusion-based methods. Furthermore, it is trained to perform speech infilling task by conditioning on both past and future audio contexts with text transcript, making Voicebox a versatile platform for various speech tasks, including zero-shot TTS synthesis, speech editing, transient noise removal, and style conversion.

On the other hand, VoiceCraft is a Transformer-based NCLM that utilizes autoregressive conditioning on bidirectional context to infill the masked neural speech codec tokens. VoiceCraft is built upon a two-step token rearrangement procedure, comprising a causal masking step and a delayed stacking step. The causal masking technique enables autoregressive generation with bidirectional context, and the combination with delayed stacking facilitates efficient multi-codebook modeling.

However, both Voicebox and VoiceCraft have their drawbacks. Voicebox generates waveforms using a HiFi-GAN vocoder with Mel spectrograms as input and is trained on a large-scale, clean audiobook corpus, LibriLight (Kahn et al., 2020). These two factors result in Voicebox **not being good at generating speech with background audio.**

On the other hand, VoiceCraft addresses the issue by training on GigaSpeech (Chen et al., 2021), which includes diverse audio conditions from sources like audiobooks, podcasts, and YouTube. Additionally, it uses EnCodec (Défossez et al., 2022) for tokenization and waveform recovery. However, VoiceCraft suffers from common hallucination-like problems found in large language models (LLMs). These issues cause VoiceCraft to **accidentally generate speech with unintended long silences, slow speaking pace, or missing or repeated words**. This will be discussed in detail in the experiment section, with examples provided in the Appendix.

To address the aforementioned issues, we propose VoiceNoNG, which aims to combine the strengths of both Voicebox and VoiceCraft. To avoid hallucination problems and enable the generation of speech with background audio, VoiceNoNG utilizes a **latent** flow-matching framework.

For the acoustic features, we use the Descript Audio Codec (DAC) (Kumar et al., 2024) instead of Mel-spectrograms. Mel-spectrogram is primarily designed based on how humans perceive speech signals, making it less suitable for modeling other audio types (e.g., noise, music, etc.). Besides being trained on a diverse range of audio sources, another reason for using DAC is its ability to compress the features into a discrete latent space through a vector quantizer (VQ). This **provides additional robustness against minor prediction errors from the editing model** (Fu et al., 2024). A small error, as long as it is not larger than the token decision boundary, will still be mapped to the correct token during VQ. This will be verified in section 3.1.6.

In addition to the original flow-matching loss used in Voicebox, we apply a cross-entropy (CE) loss to further enhance our model's codec prediction accuracy after VQ. Since the codec token selection is based on the distance between the prediction and the candidates in the dictionary $C$, we can formulate this process as a probability distribution using the softmax operation. Consequently, the CE loss $L_{ce}$ can be calculated as follows:

$$L_{ce} = -\frac{1}{|M|} \sum_{t \in M} log\left(\frac{exp(-||(S(X)_t - Z_t||_2)}{\sum_{v=1}^{V} exp(-||(S(X)_t - C_v||_2)}\right) \tag{1}$$

where $M$ represents the masked region, $S(.)$ denotes our speech editing model with input features $X$, $Z$ is the original codec token before masking, and $V$ is the size of the dictionary. This loss function aims to minimize the distance between the prediction and the correct token while maximizing the distance to other tokens. Similar to VoiceCraft, we use GigaSpeech as the training corpus, as its diverse audio conditions enable our model to perform background-preserving speech editing.

# 3 EXPERIMENTS

## 3.1 SPEECH EDITING

### 3.1.1 SPEECH EDITING MODEL SETUP

Our proposed VoiceNoNG combines the strengths of both Voicebox and VoiceCraft, enabling our model to perform background-preserving speech edits without the hallucination issues present in VoiceCraft. To verify that our modifications lead to the desired improvements, we compared our speech editing model with four other models: two different sizes of VoiceCraft (330M, 880M), and two Voicebox models using Mel-spectrograms as the acoustic feature, trained on LibriLight and GigaSpeech respectively. The results for VoiceCraft are based on the official code provided by the authors. Since no code and model checkpoints are available for Voicebox, we reproduced the results. Our Voicebox architecture settings are identical to those reported in the original Voicebox paper, which features 330M parameters.

To delve into more details, DAC comprises three modules: an encoder that encodes the input waveform into a fixed-dimension vector sequence (pre-quantization feature), a residual VQ module that discretizes the encoded representations into tokens, and a decoder that decodes the quantized representations (post-quantization feature) back into the waveform. In our experiments, we use the pre-quantization feature as our acoustic feature and apply a CE loss calculated against the ground-truth quantized token labels to enhance the token prediction accuracy of VoiceNoNG.

To demonstrate the impact of this setting, we also conduct an ablation study to highlight the benefits of modeling pre-quantization features and incorporating the CE loss.

### 3.1.2 TRAINING AND TEST SETS FOR SPEECH EDITING

Unless otherwise noted, the training data used in our experiment is GigaSpeech. For evaluating different speech editing models, we utilize the RealEdit dataset (Peng et al., 2024), a pioneering benchmark that features 310 manually-crafted examples reflecting real-world editing scenarios. This dataset includes tuples of original audio, original transcript, and edited transcript, meticulously

crafted to ensure grammatical correctness and semantic coherence. RealEdit consists of 100 utterances from the LibriTTS dataset (Zen et al., 2019), 100 from YouTube (sourced from the GigaSpeech test set (Chen et al., 2021)), and 110 from the Spotify Podcast dataset (Clifton et al., 2020). The utterances range from 5 to 12 seconds in duration and encompass a diverse array of content, accents, speaking styles, and recording conditions. The edits span various types—insertions, deletions, and substitutions—across different lengths, from short (1-2 words) to long (7-12 words), with single or multiple spans. This diversity makes RealEdit more challenging compared to other speech synthesis evaluation datasets. By incorporating a wider array of editing scenarios and audio sources, RealEdit provides a comprehensive benchmark for evaluating the performance and practicality of speech editing models in diverse, real-world contexts.

For each speech editing model, we generate edited speech for every tuple of original audio, original transcripts, and edited transcripts from the RealEdit dataset. We then evaluate these generated speeches using the metrics introduced in the following subsections to compare the performance of different models. During the objective evaluation, speech was generated with five different random seeds, and we report the mean scores along with the standard deviation (shown in parentheses).

### 3.1.3 OBJECTIVE EVALUATION FOR SPEECH INTELLIGIBILITY: WER

WER is an appropriate metric for evaluating TTS and speech editing since it provides a quantitative assessment of how accurately a system converts text into speech. Table 1 displays the WER results for various speech editing models using Whisper-large-v3 (Radford et al., 2023) as the ASR model. The WER is computed by comparing the transcribed text to the target transcript provided by the RealEdit datasets. Before calculating the WER, both texts were normalized using the whisper_normalizer [1].

As indicated in the table, all flow-matching-based methods achieve lower WER than VoiceCraft, which is consistent with the findings presented in the VoiceCraft paper (Peng et al., 2024). A closer examination reveals that this is primarily due to the hallucination problem associated with VoiceCraft. Neekhara et al. (2024) also highlights that LLM-based TTS models lack robustness, often producing outputs with repeating words, missing words, and misaligned speech, particularly when the text includes multiple instances of the same token. Examples of this hallucination issue can be found in the Appendix, specifically in Tables 4 and 5. Additionally, the results demonstrate that when the training corpus for Voicebox is switched from LibriLight to GigaSpeech, the WER decreases across all types of audio sources, and further reducing the WER is achieved by changing the acoustic feature from Mel-spectrogram to DAC.

In our proposed models, we also perform an ablation study to illustrate the advantages of modeling **pre-quantization features** and **adding CE loss**. When the output of our flow-matching model is the pre-quantization features, it must be processed through the VQ and DAC decoder to reconstruct the waveform. As noted in Section 2, the VQ module offers **additional robustness against prediction errors made by our model**. In contrast, if the output is the post-quantization features (similar to (Shen et al., 2023)), only the DAC decoder is required for waveform reconstruction. As a result, post-quantization features yield the highest WER on average. Incorporating CE loss helps improve the token prediction accuracy of pre-quantization features, thereby reducing the WER.

It is essential to highlight that the WER results presented in Table 1 are calculated for the entire utterance, and the unmasked regions are expected to exhibit the same WER. Consequently, the WER differences among various editing models should be **considerably more pronounced in the edited regions**. Additionally, while a lower WER signifies that the generated speech is more intelligible to the ASR and contains more accurate content, it does not automatically imply that the overall quality is better.

### 3.1.4 OBJECTIVE EVALUATION FOR SPEECH QUALITY: SQUIM-(SI-SDR)

To assess the quality of the generated speech, we utilize a non-intrusive estimation of the scale-invariant signal-to-distortion ratio (SI-SDR), referred to as SQUIM-(SI-SDR) from (Kumar et al., 2023). The results of the estimated quality scores are presented in Table 2. It is evident that the SI-SDR for the original speech from LibriTTS is higher than that of the speech from YouTube and Spotify. This finding corresponds with the characteristics of the datasets, as the speech from

---

[1] https://github.com/kurianbenoy/whisper_normalizer

Table 1: WER evaluation of speech editing on the RealEdit dataset. An asterisk (*) indicates the reproduced model.

| | WER (%) | | | |
|---|---|---|---|---|
| | LibriTTS | YouTube | Spotify | Total |
| Original | 1.84 | 6.13 | 5.87 | 4.65 |
| VoiceCraft (330M) | 3.72 (0.13) | 7.41 (0.38) | 4.63 (0.14) | 5.23 (0.16) |
| VoiceCraft (830M) | 3.77 (0.41) | 7.36 (0.35) | 5.43 (0.33) | 5.52 (0.19) |
| Voicebox* (Libri, Mel) | 3.64 (0.19) | 6.26 (0.19) | 5.45 (0.19) | 5.13 (0.08) |
| Voicebox* (Giga, Mel) | 3.48 (0.16) | 6.03 (0.16) | 5.36 (0.13) | 4.97 (0.12) |
| Proposed VoiceNoNG | **2.82** (0.20) | 5.84 (0.21) | **4.92** (0.21) | **4.54** (0.14) |
| Post-quantization | 3.16 (0.19) | **5.59** (0.24) | 5.38 (0.22) | 4.73 (0.18) |
| No CE loss | 2.94 (0.23) | 5.81 (0.18) | 5.07 (0.13) | 4.62 (0.09) |

Table 2: SQUIM-(SI-SDR) evaluation of speech editing on the RealEdit dataset. An asterisk (*) indicates the reproduced model. We highlight the LibriTTS case in bold, as the scores for YouTube and Spotify provide less clear insights.

| | SI-SDR | | | |
|---|---|---|---|---|
| | **LibriTTS** | YouTube | Spotify | Total |
| Original | 23.94 | 19.55 | 19.22 | 20.85 |
| VoiceCraft (330M) | 22.22 (0.08) | 19.97 (0.08) | 20.15 (0.16) | 20.76 (0.04) |
| VoiceCraft (830M) | 22.53 (0.07) | 20.04 (0.06) | 20.26 (0.13) | 20.92 (0.08) |
| Voicebox* (Libri, Mel) | 19.49 (0.13) | 17.96 (0.16) | 16.32 (0.10) | 17.88 (0.03) |
| Voicebox* (Giga, Mel) | 18.10 (0.41) | 16.86 (0.32) | 15.83 (0.17) | 16.90 (0.20) |
| Proposed VoiceNoNG | 23.15 (0.09) | 19.29 (0.05) | 19.04 (0.08) | 20.44 (0.05) |
| Post-quantization | 20.80 (0.26) | 18.28 (0.11) | 17.81 (0.10) | 18.93 (0.05) |
| No CE loss | 23.36 (0.09) | 19.30 (0.09) | 18.98 (0.06) | 20.50 (0.04) |

YouTube and Spotify contains some background audio. Although the SI-SDR scores offer less clear information regarding the generated speech for YouTube and Spotify, we still provide the scores for reference.

In the case of LibriTTS, we can observe that the speech quality produced by the proposed model is the best and closely resembles the original speech. Conversely, the quality score for the speech generated by Voicebox is the lowest, likely due to the limitations of the Mel-spectrogram and the vocoder used. We also observed that if our proposed method utilized post-quantization features as learning targets, the SI-SDR would significantly decrease. Furthermore, the standard deviations for Voicebox (Giga, Mel) and post-quantization are considerably larger than those for other models. This observation aligns with our listening experience, indicating that these models are less robust and more likely to produce varying speech quality across the five random seeds. **This further validates the advantages of predicting pre-quantization features and employing the VQ module to correct minor prediction errors**.

### 3.1.5 SUBJECTIVE EVALUATION

To subjectively evaluate the performance of various speech editing methods, we conducted a listening test using a 5-point scale. Participants listened to audio files, which could either be real or edited, and rated them on a scale where 5 represented highly natural and unaltered audio, while 1 indicated a strong belief that the audio had undergone partial editing (mainly follows (Peng et al., 2024); please refer to Figure 5 in the Appendix for detailed instructions). The naturalness score encompasses factors such as smoothness, fluidity, and the absence of robotic or synthetic artifacts, all of which are essential for assessing whether the speech sounds authentic to human listeners. 8 original speeches were randomly selected from LibriTTS, YouTube, and Spotify, respectively. The corresponding

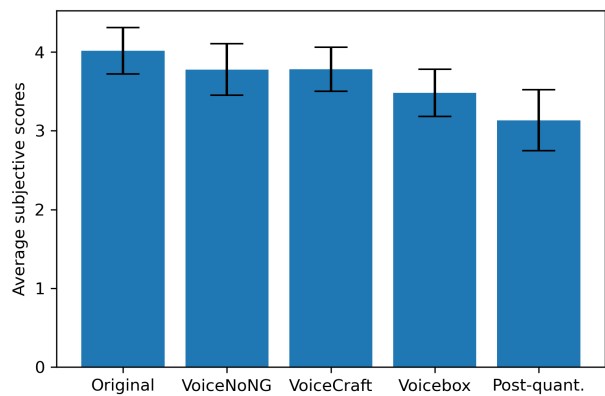

Figure 2: Subjective scores of different speech editing methods.

edited speeches from VoiceCraft, Voicebox, the proposed VoiceNoNG, and VoiceNoNG with post-quantization were then chosen, resulting in a total of (8×3)×5 = 120 utterances for each listener to evaluate. The order of the audio playback was randomized, and 15 listeners participated in the study. The experimental results (with a 95% confidence interval) are shown in Figure 2, revealing that the naturalness of speeches edited by VoiceNoNG and VoiceCraft is comparable to that of the original speech (detailed results for each subset can be found in the Appendix from Figure 6 to 8). In contrast, the performance of VoiceNoNG with post-quantization exhibits a significant decline, warranting further analysis in the next section.

### 3.1.6 ROBUSTNESS BENEFITS FROM VQ MODULE

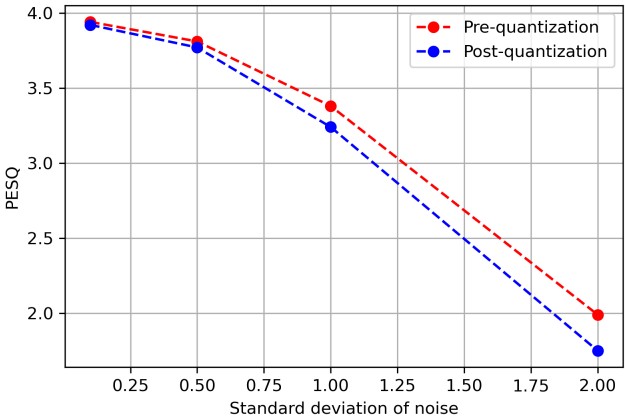

Figure 3: Relationship between varying levels of noise added to pre-quantization and post-quantization DAC features and the PESQ scores of the resulting resynthesized speech.

As shown in the previous experiments, the performance of VoiceNoNG declines when post-quantization features are used as learning targets. To validate the advantages of predicting pre-quantization features with VQ, we intentionally added Gaussian noise to these features to **simulate the prediction error incurred by the editing model**. LibriTTS subset from the RealEdit dataset was chosen for this experiment. In Figure 3, we present the PESQ scores (Rix et al., 2001) for resynthesized speech generated from the noisy DAC features at varying noise levels (the noise strength is first normalized based on the dynamic range of the input features). This figure illustrates the robustness

benefits of VQ, showing an approximate improvement of 0.2 PESQ scores under moderate noise conditions.

## 3.2 DETECTION OF SPEECH EDITED BY VOICENONG

As demonstrated in the previous subjective evaluation, people find it challenging to distinguish between speech edited by the proposed VoiceNoNG and real speech. To prevent malicious uses, a reliable Deepfake detector is essential. This section explores a practical new threat within the partial spoof scenario. Given the potential for neural codecs to become a new audio format standard, the assumption that all codec-generated speech is fake may soon be unrealistic. Therefore, in the following experiments, we classified codec-resynthesized speech as a new category. The original binary classifier was extended to a ternary classifier with the labels: **real**, **resynthesized**, and **edited**. Additionally, for the audio condition, besides the original VoiceNoNG setting where non-edited segments come from the original audio, we consider a more challenging setting where non-edited segments are also resynthesized from the codec. We refer to this condition as VoiceNoNG (resyn).

### 3.2.1 DEEPFAKE SPEECH DETECTOR MODEL SETUP

We built the w2v2-detector using a model architecture similar to the one that won first place in the partially fake audio track at ADD2022 (Yi et al., 2022), which also demonstrated strong performance in our previous work. Specifically, the w2v2-detector employs a pretrained wav2vec2-base-960h model (Baevski et al., 2020) for feature extraction. We use a weighted sum of the 13 hidden states, including the output from the CNN encoder, as our input features. These features are then projected from 768 dimensions to 128 dimensions using a linear layer. The model then splits into two paths: the **frame-level** branch and the **utterance-level** branch. In the frame-level branch, the projected features pass through a linear layer to produce a 3-dimensional output representing the prediction for each frame. In the utterance-level branch, the projected features undergo attentive statistics pooling (Okabe et al., 2018) to compress the features into a single representation of the entire utterance, which is then passed through a linear layer to generate the utterance-level prediction.

### 3.2.2 TRAINING AND TEST SETS FOR DEEPFAKE DETECTOR

We used LibriLight medium (Kahn et al., 2020) as the source audio files and generated the edited speech using our proposed VoiceNoNG. The target transcripts were generated by prompting an LLM (i.e., zephyr-7B-beta (Tunstall et al., 2023)). However, the edited transcripts may not always adhere to the desired format. To address this, we used word-level Levenshtein distance (Levenshtein et al., 1966) to identify substitutions, and then randomly selected words for replacement as our method of edit manipulation.

The dataset was first categorized by speaker, prioritizing those with more audio samples for the training set, and then sequentially allocating speakers to the validation and test sets. This approach ensured that no speaker appeared in multiple sets, with the test set containing a significant number of unseen speakers, enabling us to effectively assess the detector's generalizability at the speaker level. The dataset was divided into training, validation, and test sets, comprising 106,186, 34,744, and 33,974 audio files, respectively. Additionally, the quantities of real, resynthesized, and edited speech in the original VoiceNoNG and VoiceNoNG (resyn) settings were balanced within each set.

### 3.2.3 EXPERIMENTAL RESULTS

Table 3 presents the performance of the w2v2-detector in detecting and localizing speech edits made by the proposed VoiceNoNG model. The table reveals that the utterance-level accuracy is nearly 100%, even under the VoiceNoNG (resyn) condition. However, as anticipated, the frame-level F1 score for the VoiceNoNG (resyn) condition is lower than that for VoiceNoNG. This indicates that it is more challenging for the detector to differentiate between **edited and non-edited resynthesized** segments at the frame level.

To examine how the amount of training data affects the detector's performance, we included results for different training set sizes in the table. The table indicates that approximately 40,000 samples are sufficient to train a reliable DeepFake detector. Additionally, thanks to the advantages provided by the self-supervised front-end (i.e., wav2vec2), only 5,000 examples are needed to achieve an

Table 3: Detection and localization results of speech edited by VoiceNoNG.

| # of training sample | frame F1 (%) / utterance accuracy (%) | |
| --- | --- | --- |
| | VoiceNoNG | VoiceNoNG (resyn) |
| 106,186 | 95.84 / 98.08 | 82.63 / 96.99 |
| 80,000 | 93.59 / 97.17 | 83.14 / 97.14 |
| 40,000 | 94.44 / 97.17 | 81.28 / 96.66 |
| 20,000 | 91.25 / 94.85 | 80.89 / 92.83 |
| 5,000 | 89.39 / 91.32 | 76.45 / 88.01 |
| 1,000 | 65.97 / 79.07 | 56.86 / 64.61 |
| 500 | 58.83 / 72.79 | 47.48 / 66.19 |

utterance-level accuracy of around 90%. Figure 4 presents an example of frame-level detection results for the VoiceNoNG (resyn) condition. Additional examples under different acoustic conditions can be found in the Appendix (Figure 9 to 11). Through this experiment, we discovered that while it is challenging for people to differentiate between speech edited by the proposed VoiceNoNG and real speech, the trained w2v2-detector is still capable of detecting subtle artifacts that distinguish real audio from fake.

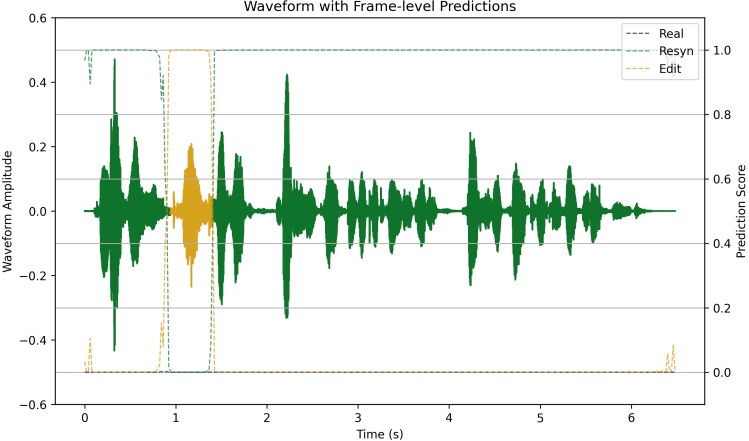

Figure 4: An example of frame-level detection for **VoiceNoNG (resyn)** condition. In the waveform, the green sections represent resynthesized speech, while the orange sections indicate edited speech. The dashed lines show the predicted scores of our detector for the three classes.

## 4    CONCLUSION

In this study, we first examine the limitations of current advanced speech editing models. Voicebox produces lower-quality speech in the presence of background audio (e.g., noise, music), while Voice-Craft struggles to accurately follow text input, a common hallucination-like issue with autoregressive models. To address these challenges, we introduce VoiceNoNG, which leverages the strengths of both models and achieves SOTA performance in both objective and subjective evaluations (we encourage readers to listen to the demos on our demo page, which shows VoiceNoNG can keep the accent and even successfully generate background music for movie editing!). We also identify the vector quantizer module in the neural codec as crucial for achieving a robust speech editing model. Lastly, to mitigate the potential risks posed by the speech editing model, we examine the performance of the Deepfake detector in a new and challenging practical scenario.

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

# Appendix

Table 4: ASR results of sampling 5 times of edited `8173_294714_000033_000000.wav` in the RealEdit dataset. Words highlighted in red indicate missing words, while those in blue denote extra words. Texts in parentheses signify simple substitution errors. Bolded texts indicate specific error types from the LLM, such as missing or repeated words caused by hallucinations.

---

**Ground truth:**

"promise that you will not ask me to borrow any money from the bank for the bail of you for mister van brandt she rejoined and i accept your help gratefully"

**VoiceCraft:**

1. "promise that you will not ask me to borrow any money **-from the bank for the bail** of you (-for +from) mister van (-brandt she +branch you) rejoined and i accept your help gratefully"

2. "promise that you will not ask me to borrow any money **-from the bank for the bail** of you (-for +from) mister van (-brandt she +branch you) rejoined and i accept your help gratefully"

3. "promise that you will not ask me to borrow any money **-from the bank for the bail** of you (-for +from) mister van (-brandt she +branch you) rejoined and i accept your help gratefully"

4. "promise that you will not ask me to borrow any money from the bank for the bail of you for mister van -brandt she **+bunny of you for mister van branch you** rejoined and i accept your help (-gratefully +greatly)"

5. "promise that you will not ask me to borrow any money from the bank for the bail of you for mister van -brandt she rejoined **+brawny of you for mister van branch you were joined** and i accept your help (-gratefully +greatly)"

**Proposed VoiceNoNG:**

1. "promise that you will not ask me to borrow any money from the bank for the (-bail +money) of you for mister van (-brandt +brant) she rejoined and i +will accept your help gratefully"

2. "promise that you will not ask me to borrow any money from the bank for the (-bail +money) of you for mister van (-brandt +brant) she rejoined and i +will accept your help gratefully"

3. "promise that you will not ask me to borrow any money from the bank for the (-bail +money) of you for mister van (-brandt +brant) she rejoined and i +will accept your help gratefully"

4. "promise that you will not ask me to borrow any money from the bank for the (-bail +money) of you for mister van (-brandt +brant) she rejoined and i +will accept your help gratefully"

5. "promise that you will not ask me to borrow any money from the bank for the (-bail +money) of you for mister van (-brandt +brant) she rejoined and i +will accept your help gratefully"

---

Table 5: ASR results of sampling 5 times of edited `YOU1000000101_S0000132.wav` in the RealEdit dataset. Words highlighted in red indicate missing words, while those in blue denote extra words. Texts in parentheses signify simple substitution errors. Bolded texts indicate specific error types from the LLM, such as missing or repeated words caused by hallucinations.

---

**Ground truth:**

"yet anytime you and i question the schemes of the dogooders or dare to dig into any of their motives were denounced as being against their humanitarian goals they say we are always against things we are never for anything"

---

**VoiceCraft:**

1. "yet anytime you and i question the schemes of the (-dogooders +dog) or dare to dig into any of their motives **-were +we are denounced as gooders we are** denounced as being against their humanitarian goals they say we are always against things we are never for anything"

2. "yet anytime you and i question the schemes of the (-dogooders +dog gooders) or dare to dig into any of their **-motives were +moo gooders we are** denounced as being against their humanitarian goals they say we are always against things we are never for anything"

3. "yet anytime you and i question the schemes of the (-dogooders +dog gooters) or dare to dig into any of their **-motives were +moogooters we are** denounced as being against their humanitarian goals they say we are always against things we are never for anything"

4. "yet anytime you and i question the schemes of the (-dogooders +dog owners) or dare to dig into any of their motives **-were +who gooders we are** denounced as being against their humanitarian goals they say we are always against things we are never for anything"

5. "yet anytime you and i question the schemes of the (-dogooders +dawgooders) or dare to dig into any of their **-motives were +motu gooders we are** denounced as being against their humanitarian goals they say we are always against things we are never for anything"

---

**Proposed VoiceNoNG:**

1. "yet anytime you and i question the schemes of the (-dogooders +dog eaters) or dare to dig into any of their (-motives were +we are) denounced as being against their humanitarian goals they say we are always against things (-we are +were) never for anything +and"

2. "yet anytime you and i question the schemes of the (-dogooders +dog gooters) or dare to dig into any of their (-motives were +we are) denounced as being against their humanitarian goals they say we are always against things (-we are +were) never for anything +and"

3. "yet anytime you and i question the schemes of the (-dogooders +dog gooters) or dare to dig into any of their (-motives were +we are) denounced as being against their humanitarian goals they say we are always against things (-we are +were) never for anything +and"

4. "yet anytime you and i question the schemes of the (-dogooders +doggoners) or dare to dig into any of their (-motives were +we are) denounced as being against their humanitarian goals they say we are always against things (-we are +were) never for anything +and"

5. "yet anytime you and i question the schemes of the (-dogooders +doggers) or dare to dig into any of their (-motives were +we are) denounced as being against their humanitarian goals they say we are always against things (-we are +were) never for anything +and"

---

**Instructions**

Some of the speeches you will listen to may have been **partially edited**. Your task is to assess the naturalness of the speech **focusing** solely on the **speaker and background audio coherence, prosody, emotion, and speech rate**. Some of the audio may come from internet videos and have background noise. Please **ignore** the noise, grammar, semantics, or other linguistic factors in your evaluation.

Please rate each audio's naturalness (i.e., human-sounding) independently from 1-5. 1 is **least** natural, and 5 is **most** natural.

Please use a headset to listen and adjust the volume level to your comfort. Each audio should only be replayed at most **twice**.

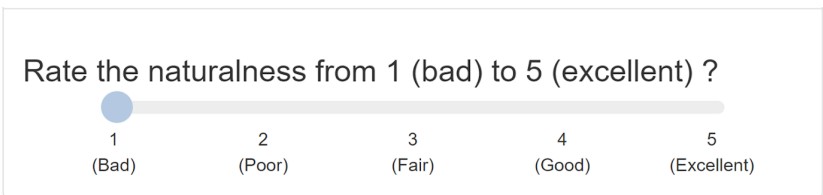

Figure 5: Instruction of the subjective evaluation.

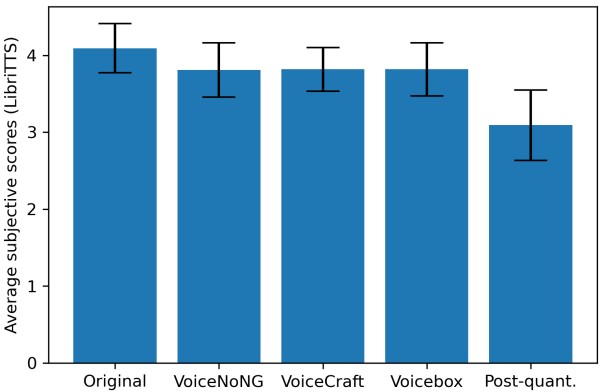

Figure 6: Subjective scores of different speech editing methods in the LibriTTS subset.

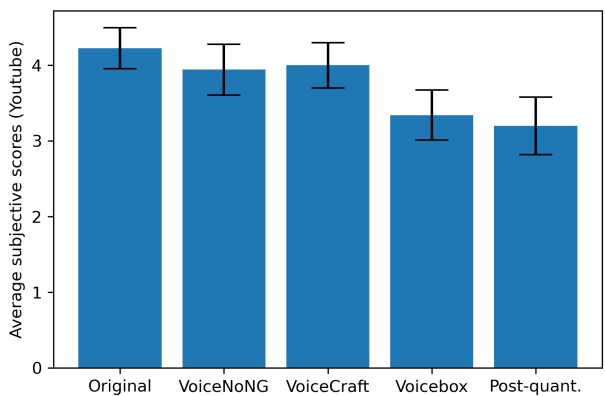

Figure 7: Subjective scores of different speech editing methods in the Youtube subset.

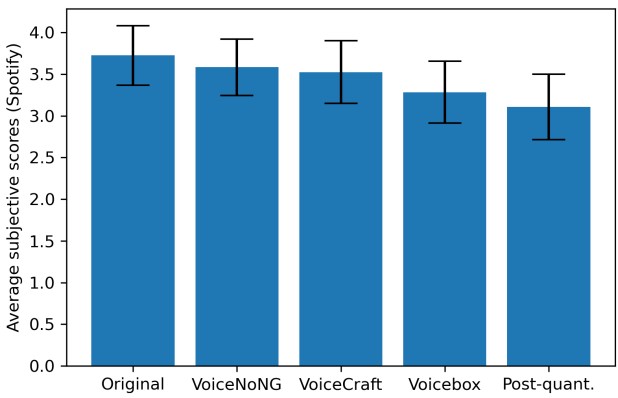

Figure 8: Subjective scores of different speech editing methods in the Spotify subset.

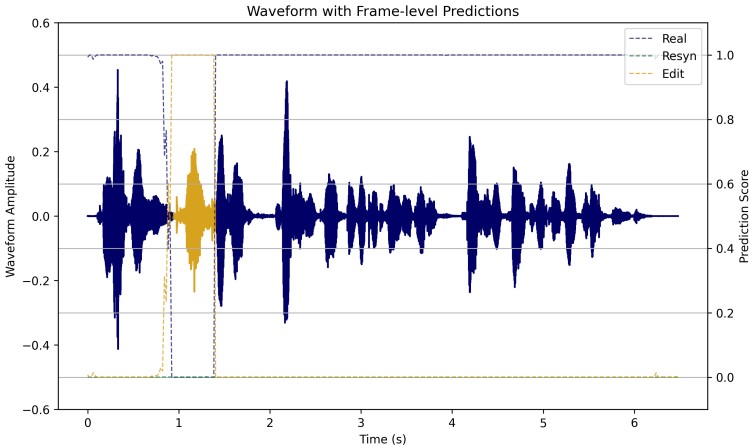

Figure 9: An example of frame-level detection for **VoiceNoNG** condition. In the waveform, the blue sections represent real speech, while the orange sections indicate edited speech.

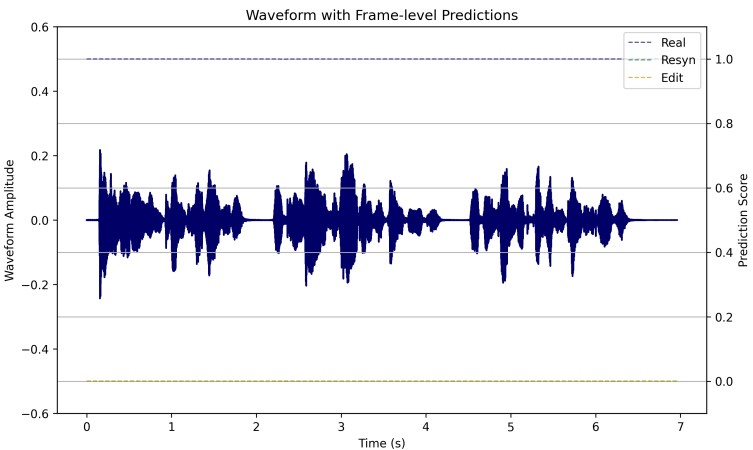

Figure 10: An example of frame-level detection for **real** condition.

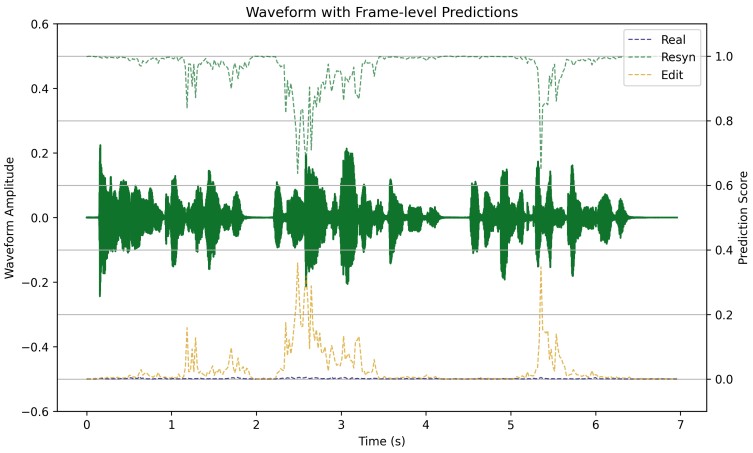

Figure 11: An example of frame-level detection for **resynthesized** condition.