# OpenReview forum: "VoiceNoNG: High-Quality Speech Editing Model without Hallucinations"
_ICLR.cc/2025/Conference — Submitted to ICLR 2025_

### Official Review · Reviewer_P8Yw · 2024-10-29

**Soundness:** 2
**Presentation:** 3
**Contribution:** 1
**Rating:** 3
**Confidence:** 4

**Summary:**

The paper introduces VoiceNoNG, a new speech editing model that combines the strengths of two existing approaches in the field: Voicebox (based on flow-matching model) and VoiceCraft (based on neural codec language models). While current speech editing models like VoiceCraft can generate high-quality speech, they suffer from hallucination issues that lead to higher word error rates, including problems like unintended silences, slow speaking pace, or missing/repeated words. VoiceNoNG addresses these issues by utilizing a latent flow-matching framework and incorporating the Descript Audio Codec (DAC) instead of traditional Mel-spectrograms as its input feature representation.
The key innovation of VoiceNoNG is its use of pre-quantization features and a vector quantizer (VQ) module, which provides additional robustness against minor prediction errors similar to quantization noise. The model also employs a cross-entropy loss to enhance codec prediction accuracy. In experimental evaluations using the RealEdit dataset, VoiceNoNG outperformed both VoiceCraft and Voicebox variants in terms of word error rate (WER) and speech quality metrics. The authors demonstrate that modeling pre-quantization features and including the VQ module are crucial for developing a robust speech editing model that can maintain high quality while avoiding hallucination issues.
Finally, the authors develop a deepfake detector by classifying edited, real and synthesized speech portion in an utterance. The proposed method is an extension of prior model by adding a new class (synthesized) into the target.

**Strengths:**

1. Well written paper with clear and concise description of the proposed method.
2. The problem statement is extremely interesting with good motivation
3. Shows that feature representation from neural codecs can be helpful for tasks like editing
4. Development of deepfake detector which is crucial given how these models can be abused by bad actors

**Weaknesses:**

1. The overall contribution is minimal with no original idea presented in the paper.
2. It appears that the authors have merely replaced the feature representation in VoiceBox model with DAC features.
3. The statements in the paper are sometimes very vague such as "This diversity makes RealEdit more challenging compared to ..." on line 222. Given that this is a scientific paper, there needs to be an explicit description of what other datasets lack which RealEdit provides.
4. Evaluation of edited speech by WER is not helpful in determining the quality/intelligibility of generation because ASR models have an implicit language model that corrects mispronunciations and even words based on context.
5. The hallucination claim is completely based on WER which is not astounding to begin with (see point 4). Further, the WER difference between Voicecraft and proposed technique is minimal except for YouTube dataset.
6. The SI-SDR difference are again very small to be meaningful across various models. This is probably not perceivable by human listeners.
7. MOS evaluation is unrealiable given there is only 3 rating per sample for a total of 10 samples per model (see ITU-T P.800 recommendations).

**Questions:**

None

---

> ### Author Response · Authors · 2024-11-19
> **Resonse to Reviewer P8Yw**
>
> ***1.& 2. The overall contribution is minimal with no original idea presented in the paper. It appears that the authors have merely replaced the feature representation in VoiceBox model with DAC features.***
>
>
> We would like to first thank the reviewer for recognizing that our paper is well-written with a clear description and strong motivation. The goal of this paper is to address the problems present in current state-of-the-art speech editing models: VoiceCraft (struggles to generate speech **accurately following** the target transcription) and Voicebox (suffers from reduced speech quality when **background audio** is present). **Identifying these issues is also one of the contributions of this paper.**
>
> Although we considered a more sophisticated framework to tackle these issues (e.g., applying a speech enhancement model to disentangle background audio from speech and performing the infilling separately), we found that our current simple framework can **already effectively** address these challenges (please listen to the demos in our demo page).
>
> Because our proposed solution is relatively elegant, we decided to focus more on the **motivation** and **experimental** parts to provide insights to the community.
>
> Although the model framework is simple, we believe this paper makes significant contributions to the related research community.
>
> - 1, We **identify the robustness problem of neural codec language models** (e.g., VoiceCraft) in speech editing.
>
> - 2, We conduct comprehensive experiments to highlight the **pros and cons** of VoiceCraft and Voicebox.
>
> - 3, Our ablation study and Figure 3 demonstrate the importance of flow-matching for **modeling pre-quantization features with the VQ module**, which can improve the robustness of models against small prediction error by implicitly converting the regression problem into a token classification task similar to NCLM.
>
> - 4, The proposed VoiceNoNG achieves **state-of-the-art performance** in both objective and subjective evaluations.
>
> - 5, Considering the potential for neural codecs to become a new audio format standard (such as mp3 format), the assumption that all codec-generated speech is fake may soon be unrealistic. Therefore, we propose a new and challenging **practical scenario for deepfake detection**, contributing to the relevant community.
>
> We kindly ask the reviewer to **listen to the demos of our proposed method and compare them with other SOTAs** (https://anonymous.4open.science/w/NoNG-8004/), and please don’t overlook the contribution of this paper due to its simple framework.
>
>
> ***3. The statements in the paper are sometimes very vague such as "This diversity makes RealEdit more challenging compared to ..." on line 222. Given that this is a scientific paper, there needs to be an explicit description of what other datasets lack which RealEdit provides.***
>
> In our original paper, the full context is: “The edits span various types—insertions, deletions, and substitutions—across different lengths, from short (1-2 words) to long (7-12 words), with single or multiple spans. This diversity makes RealEdit more challenging compared to other speech synthesis evaluation datasets.”
> As stated in the context, the diverse edit types (insertions, deletions, and substitutions) and different edit lengths (short, long) are what other datasets do not contain. We will revise the paper to make this more clear.
>
> ***4. Evaluation of edited speech by WER is not helpful in determining the quality/intelligibility of generation because ASR models have an implicit language model that corrects mispronunciations and even words based on context.***
>
> We partially **disagree** with this statement. While it is true that ASR systems may correct mispronunciations based on context, they still provide **valuable** information about the intelligibility of generated speech. Additionally, previous speech editing works, such as VoiceBox and VoiceCraft, **DO** report WER results of ASR as a **metric of intelligibility**. For instance, section 4 of VoiceBox states: “Correctness and intelligibility: This can be measured by the word error rate (WER) of the synthesized speech’s transcription with respect to the input text, which has been adopted in prior work [Wang et al., 2018]. Public automatic speech recognition (ASR) models are used for comparability.” Similarly, VoiceCraft reports WER results in their Tables 3 and 4.

---

> > ### Author Response · Authors · 2024-11-19
> > **Resonse to Reviewer P8Yw (part2)**
> >
> > ***5. The hallucination claim is completely based on WER which is not astounding to begin with (see point 4). Further, the WER difference between Voicecraft and proposed technique is minimal except for YouTube dataset.***
> >
> > We indeed identified the hallucination problem in VoiceCraft through its higher WER (despite its better quality compared to Voicebox). After further investigation, we found that VoiceCraft **cannot generate speech accurately following the target transcription.** We demonstrated two examples of such hallucinations in Tables 4 and 5 and provided the corresponding audio examples on our demo page (the link is provided in the paper abstract). You can easily access these audio examples under the section "2. Examples of attention errors (hallucinations) of VoiceCraft" on the demo page.
> >
> > Some recent papers also found this issue in the LLM-based TTS models [1][2].
> >
> > Compare the WER of VoiceCraft(830M) with the proposed VoiceNoNG: (LibriTTS) 3.77 vs 2.82, (YouTube) 7.36 vs 5.84, and (Spotify) 5.43 vs 4.92. We believe the WER difference is obvious.
> >
> > [1] Neekhara, P., Hussain, S., Ghosh, S., Li, J., Valle, R., Badlani, R., & Ginsburg, B. (2024). “Improving robustness of llm-based speech synthesis by learning monotonic alignment.” arXiv preprint arXiv:2406.17957.
> >
> > [2] Battenberg, E., Skerry-Ryan, R. J., Stanton, D., Mariooryad, S., Shannon, M., Salazar, J., & Kao, D. (2024). “Very Attentive Tacotron: Robust and Unbounded Length Generalization in Autoregressive Transformer-Based Text-to-Speech.” arXiv preprint arXiv:2410.22179.
> >
> > ***6. The SI-SDR difference are again very small to be meaningful across various models. This is probably not perceivable by human listeners.***
> >
> > As stated in the paper, we emphasize the **LibriTTS** case because the SI-SDR scores for YouTube and Spotify, which contain background audio, provide less clear meaning. For LibriTTS, the SI-SDR of Voicebox is 19.49 dB, whereas our proposed VoiceNoNG achieves 23.15 dB. This **3.66 dB** improvement is easily perceivable by human listeners. We kindly ask the reviewer to **visit our demo page**, where the quality difference between Voicebox and VoiceNoNG is evident. This difference is also reflected in our subjective listening test results shown in Figure 2.
> >
> >
> > ***7. MOS evaluation is unrealiable given there is only 3 rating per sample for a total of 10 samples per model (see ITU-T P.800 recommendations).***
> >
> > We believe there is a **misunderstanding** regarding our experiment. We have 8x3 = 24 samples per model, and each audio sample is rated by 15 listeners. For a detailed description, please refer to section 3.1.5.

---

> > > ### Author Response · Authors · 2024-11-27
> > > **Waiting for the reply**
> > >
> > > Thank you for taking the time to review our paper. We have addressed your concerns in our submitted response. As the rebuttal period is nearing its conclusion, we kindly request you to review our rebuttal and share any additional comments or concerns you may have. Thank you once again for your valuable feedback, and we are happy to answer any further questions!

---

### Official Review · Reviewer_xqLA · 2024-11-01

**Soundness:** 2
**Presentation:** 2
**Contribution:** 2
**Rating:** 5
**Confidence:** 5

**Summary:**

This paper first examines the limitations of current advanced speech editing models. Voicebox produces lower-quality speech when background audio (such as noise or music) is present, while VoiceCraft struggles to accurately follow text input, a common hallucination issue with autoregressive models. To address these challenges, the paper introduces VoiceNoNG, which leverages the advantages of both models. The authors also explore the impact of the vector quantization module in the neural codec on achieving a robust speech editing model. Finally, to mitigate potential risks posed by the speech editing model, the authors examine the performance of a deepfake detector in a new, challenging practical scenario.

**Strengths:**

The paper first discusses the advantages and limitations of two different architectures for speech editing models. It then introduces a novel speech editing model based on DAC and latent flow-matching, which can be seen as an improvement to the VoiceBox model. The improved model achieves a lower WER and enhanced speech quality. The authors also investigate the impact of the VQ module on speech editing, with findings that may extend to other application areas. Finally, the authors examine the performance of a deepfake detector in a new and challenging practical scenario, contributing to the deepfake detection community.

**Weaknesses:**

The paper presents a rather simplistic introduction to the proposed method, with much of the first two chapters focusing on popular science explanations. In the abstract, the authors state that the poor performance of the NCLM-based model is due to attention errors (hallucination phenomena), but they provide only a few examples to support this claim, lacking more extensive experiments on attention mechanisms. The paper mentions that the model combines the advantages of VoiceCraft and VoiceBox, but it seems to only merge the Codec with VoiceBox. Additionally, the authors point out that the poor performance of VoiceBox is due to the HiFi-GAN being trained on clean speech; however, the experimental section lacks a comparison with a HiFi-GAN trained on noisy speech for VoiceBox.

**Questions:**

Q1: The paper states that the poor performance of the NCLM-based model is due to attention errors (hallucination phenomena). However, the poor performance of LM-based models could also be influenced by factors such as sampling methods and codebook size. How can it be proven that the issues are specifically caused by hallucination?

Q2: In line 199, the statement "Since no code and model checkpoints are available for Voicebox, we reproduced the results" raises a question. Does this mean that you retrained Voicebox based on open-source code, or did you replicate the experimental results from Voicebox? If it is the latter, please specify which tables the data comes from, as I could not find the same data in the Voicebox paper.

Q3: The "Spotify" column in Table 1 should indicate that VoiceCraft (330M) performs the best.

Q4: Is VoiceBox and HiFi-GAN trained on GigaSpeech, and then compared with it?

---

> ### Author Response · Authors · 2024-11-19
> **Resonse to Reviewer xqLA**
>
> ***The paper presents a rather simplistic introduction to the proposed method, with much of the first two chapters focusing on popular science explanations.***
>
> The goal of this paper is to address the problems present in current state-of-the-art speech editing models: VoiceCraft (struggles to generate speech **accurately following** the target transcription) and Voicebox (suffers from reduced speech quality when **background audio** is present). **Identifying these issues is also one of the contributions of this paper.**
>
> Although we considered a more sophisticated framework to tackle these issues (e.g., applying a speech enhancement model to disentangle background audio from speech and performing the infilling separately), we found that our current simple framework can **already effectively** address these challenges (please listen to the demos in our demo page).
>
> Because our proposed solution is relatively elegant, we decided to focus more on the **motivation** and **experimental** parts to provide insights to the community.
>
> Although the model framework is simple, we believe this paper makes significant contributions to the related research community.
>
> - 1, We **identify the robustness problem of neural codec language models** (e.g., VoiceCraft) in speech editing.
>
> - 2, We conduct comprehensive experiments to highlight the **pros and cons** of VoiceCraft and Voicebox.
>
> - 3, Our ablation study and Figure 3 demonstrate the importance of flow-matching for **modeling pre-quantization features with the VQ module**, which can improve the robustness of models against small prediction error by implicitly converting the regression problem into a token classification task similar to NCLM.
>
> - 4, The proposed VoiceNoNG achieves **state-of-the-art performance** in both objective and subjective evaluations.
>
> - 5, Considering the potential for neural codecs to become a new audio format standard (such as mp3 format), the assumption that all codec-generated speech is fake may soon be unrealistic. Therefore, we propose a new and challenging **practical scenario for deepfake detection**, contributing to the relevant community.
>
> We kindly ask the reviewer to **listen to the demos of our proposed method and compare them with other SOTAs** (https://anonymous.4open.science/w/NoNG-8004/), and please don’t overlook the contribution of this paper due to its simple framework.
>
> ***In the abstract, the authors state that the poor performance of the NCLM-based model is due to attention errors (hallucination phenomena), but they provide only a few examples to support this claim, lacking more extensive experiments on attention mechanisms.***
>
> Please see our reply to Q1.
>
> ***The paper mentions that the model combines the advantages of VoiceCraft and VoiceBox, but it seems to only merge the Codec with VoiceBox.***
>
> We believe the **discrete Codec token** is crucial not only for NCLM but also for the diffusion model, particularly the VQ module in the Codec. The VQ module can implicitly convert the **regression** problem into a token **classification** task, enhancing **robustness against prediction errors** made by the editing model, as shown in Figure 3.
>
>
> ***Additionally, the authors point out that the poor performance of VoiceBox is due to the HiFi-GAN being trained on clean speech; however, the experimental section lacks a comparison with a HiFi-GAN trained on noisy speech for VoiceBox.***
>
> Please see our reply to Q4.

---

> > ### Author Response · Authors · 2024-11-19
> > **Resonse to Reviewer xqLA (part2)**
> >
> > ***Q1: The paper states that the poor performance of the NCLM-based model is due to attention errors (hallucination phenomena). However, the poor performance of LM-based models could also be influenced by factors such as sampling methods and codebook size. How can it be proven that the issues are specifically caused by hallucination?***
> >
> >
> > Thank you for highlighting this! The NCLM-based model performs well in terms of speech quality, indicating that the codebook size is sufficiently large. However, its limitation lies in the inability to **generate speech that accurately matches the target transcription.** The phenomenon of hallucinations in LLM-based TTS is a recently identified issue (please refer to [1, 2] for further details; we also cited [1] in our paper). [1] mentioned that “LLM-based TTS models are not robust as the generated output can contain repeating words, missing words and mis-aligned speech (referred to as hallucinations or attention errors), especially when the text contains multiple occurrences of the same token.”
> >
> > Although the authors of the Voicecraft paper also partially observed this issue and discussed it in Section 7: Limitations: “First and foremost is the long silence and scratching sound that occasionally occur during generation. Although in this work, we overcome it with sampling multiple utterances and selecting the shorter ones, more elegant and efficient methods are needed.”, they seem unaware that this issue may be related to the higher WER (Table 4 in Voicecraft paper).
> >
> > We demonstrated two examples of such hallucinations in Tables 4 and 5 and provided the corresponding audio examples on our demo page (the link is provided in the paper abstract). You can easily access these audio examples under the section "2. Examples of attention errors (hallucinations) of VoiceCraft" on the demo page.
> >
> >
> > [1] Neekhara, P., Hussain, S., Ghosh, S., Li, J., Valle, R., Badlani, R., & Ginsburg, B. (2024). Improving robustness of llm-based speech synthesis by learning monotonic alignment. arXiv preprint arXiv:2406.17957.
> >
> > [2] Battenberg, E., Skerry-Ryan, R. J., Stanton, D., Mariooryad, S., Shannon, M., Salazar, J., & Kao, D. (2024). Very Attentive Tacotron: Robust and Unbounded Length Generalization in Autoregressive Transformer-Based Text-to-Speech. arXiv preprint arXiv:2410.22179.
> >
> > ***Q2: In line 199, the statement "Since no code and model checkpoints are available for Voicebox, we reproduced the results" raises a question. Does this mean that you retrained Voicebox based on open-source code, or did you replicate the experimental results from Voicebox? If it is the latter, please specify which tables the data comes from, as I could not find the same data in the Voicebox paper.***
> >
> > We retrained Voicebox by ourselves with a training pipeline similar to the proposed VoiceNoNG.
> >
> > ***Q3: The "Spotify" column in Table 1 should indicate that VoiceCraft (330M) performs the best.***
> >
> > Thank you for pointing this out! We have revised this issue.
> >
> > ***Q4: Is VoiceBox and HiFi-GAN trained on GigaSpeech, and then compared with it?***
> >
> > Although we don't have the results for HiFi-GAN trained on GigaSpeech, we believe the performance may not be as robust as our VoiceNoNG, as HiFi-GAN lacks a VQ module. The VQ module helps implicitly transform the **regression** problem into a token **classification** task, which enhances robustness against **prediction errors** introduced by the editing model, as demonstrated in Figure 3.
> >
> >
> > We believe the results of 'Post-quantization' can provide some information for HiFi-GAN trained on GigaSpeech. Directly replacing the HiFi-GAN vocoder with the DAC decoder is not an optimal solution. Although Tables 1 and 2 show that using 'Post-quantization' with DAC results in better WER (4.73 vs. 4.97) and speech quality (18.93 dB vs. 16.90 dB) compared to Voicebox(Giga,Mel), applying **Pre-quantization with the VQ module and CE loss** achieves the best performance (WER: 4.54, speech quality: 20.44 dB).

---

### Official Review · Reviewer_vP9D · 2024-11-04

**Soundness:** 3
**Presentation:** 1
**Contribution:** 2
**Rating:** 3
**Confidence:** 5

**Summary:**

This paper introduces a new speech editing model called VoiceNoNG, which is based on VoiceBox and has been improved in two ways. On the one hand, the target is replaced with DAC features, and on the other hand, it is trained on the gigaspeech dataset to improve the effectiveness of speech editing, especially in scenarios with background noise. The paper also proposes a new deepfake speech detection method that considers reconstructed real speech.

**Strengths:**

1. The method proposed in the article has its rationality, enhancing the effect of speech editing by predicting high-dimensional features instead of Mel features. Since DAC features are better for information compression, replacing Mel is a good solution.
2. The article proposes more practical deepfake detection metrics and corresponding models, which have certain practical value.

**Weaknesses:**

1. The pre-quantization feature by predicting DAC is a variant of Latent Diffusion, which has been widely proven to be effective. So the solution is not novel. The novelty comes from the additional CE loss.
2. The new method proposed in the article, in my opinion, should be experimented on both TTS and speech editing, and compared with the VoiceBox model.
3. This article describes the VoiceCraft model extensively, just to show that using gigaspeech data sets can enhance the effect of speech editing with background sound. In my opinion, data selection is an engineering problem and should not be presented as a major contribution. The claim (These two factors result in Voicebox not being good at generating speech with background audio.) in Line 154 is not scientific, This drawback is due to training data, not Voicebox's drawback.
4. Rough writing, lack of model or flow chart. There is also a problem with the organization of the article. For example, the drawback of VoiceBox and VoiceCraft (Line 152 and Line 156) should not be placed in the proposed VoiceNoNG section. The ablation description (Line 248) should not be included in the WER metric. The article is in an unpolished or even unfinished state.

**Questions:**

See above. In my opinion, the contribution of the article is limited or not fully explored. And more importantly, the rough writing makes it within an unpolished or even unfinished state.

---

> ### Author Response · Authors · 2024-11-19
> **Resonse to Reviewer vP9D**
>
> ***The pre-quantization feature by predicting DAC is a variant of Latent Diffusion, which has been widely proven to be effective. So the solution is not novel....***
>
> The main technical contribution we aim to highlight is that applying a VQ module provides **additional robustness**, as demonstrated in Section 3.1.6. We argue that VQ transforms the **regression** problem into a token **classification** task (similar to NCLM). As a result, small prediction errors—provided they do not exceed the token decision boundary—are still mapped to the correct token after VQ. **While simple, we believe this observation will be valuable to the broader research community.**
>
>
> The goal of this paper is to address the problems present in current state-of-the-art speech editing models: VoiceCraft (struggles to generate speech **accurately following** the target transcription) and Voicebox (suffers from reduced speech quality when **background audio** is present). **Identifying these issues is also one of the contributions of this paper.**
>
> Although we considered a more sophisticated framework to tackle these issues (e.g., applying a speech enhancement model to disentangle background audio from speech and performing the infilling separately), we found that our current simple framework can **already effectively** address these challenges (please listen to the demos in our demo page).
>
> Because our proposed solution is relatively elegant, we decided to focus more on the **motivation** and **experimental** parts to provide insights to the community.
>
> Although the model framework is simple, we believe this paper makes significant contributions to the related research community.
>
> - 1, We **identify the robustness problem of neural codec language models** (e.g., VoiceCraft) in speech editing.
>
> - 2, We conduct comprehensive experiments to highlight the **pros and cons** of VoiceCraft and Voicebox.
>
> - 3, Our ablation study and Figure 3 demonstrate the importance of flow-matching for **modeling pre-quantization features with the VQ module**, which can improve the robustness of models against small prediction error by implicitly converting the regression problem into a token classification task similar to NCLM.
>
> - 4, The proposed VoiceNoNG achieves **state-of-the-art performance** in both objective and subjective evaluations.
>
> - 5, Considering the potential for neural codecs to become a new audio format standard (such as mp3 format), the assumption that all codec-generated speech is fake may soon be unrealistic. Therefore, we propose a new and challenging **practical scenario for deepfake detection**, contributing to the relevant community.
>
> We kindly ask the reviewer to **listen to the demos of our proposed method and compare them with other SOTAs** (https://anonymous.4open.science/w/NoNG-8004/), and please don’t overlook the contribution of this paper due to its simple framework.
>
>
>
>
>
> ***The new method proposed in the article, in my opinion, should be experimented on both TTS and speech editing, and compared with the VoiceBox model.***
>
> Thank you for your suggestion! We will consider exploring the TTS setting as part of our future work, as the proposed VoiceNoNG can also function as a TTS model. However, we believe speech editing presents a more challenging task in terms of content coherence. For the edited segment to sound natural, the speaker characteristics and background audio (e.g., noise, music, etc.) must remain consistent with the surrounding context.
>
>
> ***This article describes the VoiceCraft model extensively, just to show that using gigaspeech data sets can enhance the effect of speech editing with background sound. In my opinion....***
>
>
> For a fair comparison, we also report the Voicebox results trained on **GigaSpeech**, as shown in Tables 1 and 2. Despite being trained on GigaSpeech, Voicebox still exhibits worse WER and speech quality compared to the proposed VoiceNoNG. This highlights a key drawback of Voicebox: its model framework, which relies on **mel-spectrograms and HiFi-GAN**.
>
> Directly replacing the HiFi-GAN vocoder with the DAC decoder is also **NOT** an optimal solution. Although Tables 1 and 2 show that using 'Post-quantization' with DAC results in better WER (4.73 vs. 4.97) and speech quality (18.93 dB vs. 16.90 dB) compared to Voicebox*(Giga,Mel), applying **Pre-quantization with the VQ module and CE loss** achieves the best performance (WER: 4.54, speech quality: 20.44 dB).
>
>
> ***Rough writing, lack of model or flow chart. There is also a problem with the organization of the article...***
>
> We apologize for giving you this impression. In fact, we carefully polished the paper before submission. Given that our proposed solution is relatively simple, we chose to emphasize the **motivation** and **experimental** sections to provide valuable insights to the community. Regarding the organization of the paper, we will follow your suggestion and make revisions.

---

### Official Review · Reviewer_5Wua · 2024-11-05

**Soundness:** 2
**Presentation:** 2
**Contribution:** 2
**Rating:** 3
**Confidence:** 4

**Summary:**

This paper presents a speech generation model for speech editing that incorporates future context to ensure smooth and seamless transitions. The model is built on the Voicebox architecture, but with a modification that replaces the intermediate acoustic feature mel-spectrogram with the continuous hidden features of the neural codec model DAC. The author demonstrates that this modification leads to some performance improvements, particularly in terms of Word Error Rate (WER).

**Strengths:**

The authors compare the proposed method against two baselines, VoiceCraft and VoiceBox, representing autoregressive (AR) and non-autoregressive (NAR) models, respectively. The results demonstrate a performance improvement. Additionally, the authors conduct an ablation study that shows the effectiveness of the proposed approach.

**Weaknesses:**

There are several fundamental issues with this work:
- **Unbalanced Structure**: The paper's structure is uneven, with the first three pages primarily dedicated to background and related work, while the proposed method is introduced briefly in a single paragraph at the end of Section 2. This creates an imbalance that detracts from the focus on the contributions of the work.
- **Questionable Claim About Voicebox Performance**: In Section 2, the authors suggest that the poor performance of Voicebox in generating speech with background audio is due to its use of the HiFi-GAN vocoder and mel-spectrogram. However, this assertion could be problematic. HiFi-GAN with mel-spectrograms has been demonstrated to effectively generate a wide variety of sounds, including music and singing voices. This raises doubts about the validity of the paper’s motivation, as it seems based on an incorrect premise.
- **Prior Work Overlooked**: The introduction of cross-entropy loss as an auxiliary loss function for diffusion models has already been proposed in NaturalSpeech 2. The authors should clearly acknowledge this prior work in Section 2, as this is not a novel contribution and may lead to some confusion regarding the originality of the approach.
- **Previous Work on Codec Embeddings**: The use of diffusion models to generate codec embeddings was already explored in NaturalSpeech 2. Although the authors claim that performance differences arise from using embeddings from different layers (before versus after quantization), the novelty of this contribution appears limited and unlikely to significantly expand the current body of knowledge in this area.
- **Concerns About WER Evaluation**: Although it is not explicitly stated, it seems that the entire utterance is passed through the vocoder or codec decoder to generate the waveform. If this is the case, the claim in Section 3.1.3 that "the unmasked regions are expected to exhibit the same WER, and thus the WER differences among various editing models should be considerably more pronounced in the edited regions" seems problematic. This is because the WER would be heavily influenced by the choice of vocoder or codec decoder. In Table 1, since different models use different vocoders and codec decoders, the observed WER gap may primarily reflect differences in the vocoders rather than the models themselves. As such, the current experiments do not adequately support this argument.
- **Reference missing**:
    - Seed-TTS: A Family of High-Quality Versatile Speech Generation Models
    - UniCATS: A Unified Context-Aware Text-to-Speech Framework with Contextual VQ-Diffusion and Vocoding
    - E1 TTS: Simple and Fast Non-Autoregressive TTS

**Questions:**

The questions are listed above in the weaknesses.

---

> ### Author Response · Authors · 2024-11-19
> **Resonse to Reviewer 5Wua**
>
> ***1. Unbalanced Structure:***
>
>
> Thank you for pointing this out! The goal of this paper is to address the problems present in current state-of-the-art speech editing models: VoiceCraft (struggles to generate speech **accurately following** the target transcription) and Voicebox (suffers from reduced speech quality when **background audio** is present). **Identifying these issues is also one of the contributions of this paper.**
>
> Although we considered a more sophisticated framework to tackle these issues (e.g., applying a speech enhancement model to disentangle background audio from speech and performing the infilling separately), we found that our current simple framework can **already effectively** address these challenges (please listen to the demos in our demo page).
>
> Because our proposed solution is relatively elegant, we decided to focus more on the **motivation** and **experimental** parts to provide insights to the community.
>
> Although the model framework is simple, we believe this paper makes significant contributions to the related research community.
>
> - 1, We **identify the robustness problem of neural codec language models** (e.g., VoiceCraft) in speech editing.
>
> - 2, We conduct comprehensive experiments to highlight the **pros and cons** of VoiceCraft and Voicebox.
>
> - 3, Our ablation study and Figure 3 demonstrate the importance of flow-matching for **modeling pre-quantization features with the VQ module**, which can improve the robustness of models against small prediction error by implicitly converting the regression problem into a token classification task similar to NCLM.
>
> - 4, The proposed VoiceNoNG achieves **state-of-the-art performance** in both objective and subjective evaluations.
>
> - 5, Considering the potential for neural codecs to become a new audio format standard (such as mp3 format), the assumption that all codec-generated speech is fake may soon be unrealistic. Therefore, we propose a new and challenging **practical scenario for deepfake detection**, contributing to the relevant community.
>
> We kindly ask the reviewer to **listen to the demos of our proposed method and compare them with other SOTAs** (https://anonymous.4open.science/w/NoNG-8004/), and please don’t overlook the contribution of this paper due to its simple framework.
>
>
>
> ***2. Questionable Claim About Voicebox Performance:***
>
> Several research papers [1-2] have indicated that **“HiFi-GAN does not generalize well to non-speech audio such as sound or music”** [1]. Reviewers can listen to some distorted examples of HiFi-GAN generated music at this link: https://bigvgan-demo.github.io/. In fact, our subjective listening test (Figure 6) shows that Voicebox has comparable quality to VoiceCraft and the proposed VoiceNoNG under **clean** conditions (LibriTTS). However, as shown in Figures 7 and 8, when there is **background audio** (YouTube and Spotify), the speech quality from Voicebox is worse than that of VoiceCraft and the proposed VoiceNoNG .
>
> [1] Vyas, A., Shi, B., Le, M., Tjandra, A., Wu, Y. C., Guo, B., ... & Hsu, W. N. (2023). “Audiobox: Unified audio generation with natural language prompts.” arXiv preprint arXiv:2312.15821.
>
> [2] S.-g. Lee, W. Ping, B. Ginsburg, B. Catanzaro, and S. Yoon. “Bigvgan: A universal neural vocoder with large-scale training.” arXiv preprint arXiv:2206.04658, 2022.
>
>
>
> ***3. Prior Work Overlooked:***
>
> Thank you for pointing this out, we will cite and discuss NaturalSpeech 2’s method when introducing equation (1).

---

> ### Author Response · Authors · 2024-11-19
> **Resonse to Reviewer 5Wua (part2)**
>
> ***4.Previous Work on Codec Embeddings:***
>
>
> Yes, NaturalSpeech 2 has explored generating codec embeddings using diffusion models.
>
> However, as described in our section 3.1.3: “As noted in Section 2, the VQ module offers **additional robustness against prediction errors made by our model**. In contrast, if the output is the post-quantization features (similar to NaturalSpeech 2), only the DAC decoder is required for waveform reconstruction.”
>
> We aim to highlight to the community that **applying a VQ module can provide extra robustness benefits**, as verified in our section 3.1.6. Unlike NaturalSpeech 2, which directly models quantized vectors without applying a VQ during inference (see Figure 2 in their paper), we argue that VQ can convert the **regression** problem into a token **classification** (similar to NCLM). Hence, a small prediction error, as long as it is not larger than the token decision boundary, will still be mapped to the correct token after VQ.
> We believe these insights can help the community build more robust diffusion models.
>
>
> ***5.Concerns About WER Evaluation:***
>
> No, we followed the Voicebox approach, where unmasked regions are directly copied from the original waveform. As mentioned in Section 3.2: 'Additionally, for the audio condition, besides the original VoiceNoNG setting where non-edited segments come from the original audio, we consider a more challenging setting where non-edited segments are also resynthesized from the codec. We refer to this condition as VoiceNoNG (resyn).' The scenario where the **entire utterance is passed through the vocoder or codec decoder to generate the waveform** is only considered in the study of detecting edited speech.
>
>
> ***6.Reference missing:***
>
>
> Thanks for sharing these papers. We will cite and discuss these papers in section 1 (Introduction).

---

### Meta-Review · Area_Chair_aFdF · 2024-12-21

**Metareview:**

**Paper Summary:**

This paper proposes an adaptation of the Voicebox architecture for speech editing, replacing the intermediate Mel-spectrogram representation with continuous pre-quantized neural audio codec features. Experiments show that this modification outperforms the baseline Voicebox (diffusion) as well as the VoiceCraft (autoregressive model over quantized neural audio codec tokens) editing models on the RealEdit dataset.

**Strengths:**

Reviewers agree that the change of feature representation (Mel-spectrogram -> neural audio codec) is interesting and well-motivated.

**Weaknesses:**

All reviewers raised significant issues with the exposition of this paper. "The paper's structure is uneven" (5Wua). "The article is in an unpolished or even unfinished state" (vP9D). "The paper presents a rather simplistic introduction to the proposed method" (xqLA). "The statements in the paper are sometimes very vague" (P8Yw). Reviewers also raise significant concerns about the experimental claims with respect to training datasets (vP9D, xqLA) and evaluation metrics (5Wua, P8Yw).

**Additional Comments On Reviewer Discussion:**

The authors have responded extensively to the reviewers' comments. On the question of exposition in particular: the authors have largely disputed reviewer criticism, rather than consider how best to improve the organization and claims of the paper. I broadly agree with the reviewer assessments about both structural issues with the paper, and specific concerns about unscientific claims. I urge the authors to take the reviewer feedback constructively.

---

### Decision · Program_Chairs · 2025-01-22

Reject